# Effects of Pomegranate Seed Oil on Lower Extremity Ischemia-Reperfusion Damage: Insights into Oxidative Stress, Inflammation, and Cell Death

**DOI:** 10.3390/medicina61020212

**Published:** 2025-01-24

**Authors:** Ümmü Gülşen Bozok, Aydan İremnur Ergörün, Ayşegül Küçük, Zeynep Yığman, Ali Doğan Dursun, Mustafa Arslan

**Affiliations:** 1Department of Physiology, Faculty of Medicine, Ankara Medipol University, Ankara 06230, Turkey; gulsenozsoy07@gmail.com; 2Department of Anesthesiology and Reamination, Faculty of Medicine, Gazi University, Ankara 06510, Turkey; aydaniremnurergorun@gazi.edu.tr; 3Department of Physiology, Faculty of Medicine, Kutahya Health Sciences University, Kutahya 43020, Turkey; kucukaysegul@hotmail.com; 4Department of Histology and Embryology, Faculty of Medicine, Gazi University, Ankara 06500, Turkey; zeynepyigman@gazi.edu.tr; 5Neuroscience and Neurotechnology Center of Excellence, NÖROM, Gazi University, Ankara 06560, Turkey; 6Department of Physiology, Faculty of Medicine, Atılım University, Ankara 06830, Turkey; ali.dursun@atilim.edu.tr; 7Vocational School of Health Services, Atilim University, Cankaya, Ankara 06805, Turkey; 8Home Care Services, Medicana International Ankara Hospital, Cankaya, Ankara 06520, Turkey; 9Application and Research Centre for Life Sciences, Gazi University, Ankara 06830, Turkey; 10Centre for Laboratory Animal Breeding and Experimental Research (GÜDAM), Gazi University, Ankara 06560, Turkey

**Keywords:** apoptosis, inflammation, ischemia–reperfusion, lower extremity, NF-κB, oxidative stress, pomegranate seed oil

## Abstract

*Aim*: This study sought to clarify the therapeutic benefits and mechanisms of action of pomegranate seed oil (PSO) in instances of ischemia–reperfusion (IR) damage in the lower extremities. *Materials and Methods*: The sample size was determined, then 32 rats were randomly allocated to four groups: Control (C), ischemia–reperfusion (IR), low-dose PSO (IR + LD, 0.15 mL/kg), and high-dose PSO (IR + HD, 0.30 mL/kg). The ischemia model in the IR group was established by occluding the infrarenal aorta for 120 min. Prior to reperfusion, PSO was delivered to the IR + LD and IR + HD groups at doses of 0.15 mL/kg and 0.30 mL/kg, respectively, followed by a 120 min reperfusion period. Subsequently, blood and tissue specimens were obtained. Statistical investigation was executed utilizing Statistical Package for the Social Sciences version 20.0 (SPSS, IBM Corp., Armonk, NY, USA). *Results*: Biochemical tests revealed significant variations in total antioxidant level (TAS), total oxidant level (TOS), and the oxidative stress index (OSI) across the groups (*p* < 0.0001). The IR group had elevated TOS and OSI levels, whereas PSO therapy resulted in a reduction in these values (*p* < 0.05). As opposed to the IR group, TASs were higher in the PSO-treated groups. Histopathological analysis demonstrated muscle fiber degeneration, interstitial edema, and the infiltration of cells associated with inflammation in the IR group, with analogous results noted in the PSO treatment groups. Immunohistochemical analysis revealed that the expressions of Tumor Necrosis Factor-alpha (TNF-α), Nuclear Factor kappa B (NF-κB), cytochrome C (CYT C), and caspase 3 (CASP3) were elevated in the IR group, while PSO treatment diminished these markers and attenuated inflammation and apoptosis (*p* < 0.05). The findings demonstrate that PSO has a dose-dependent impact on IR injury. *Discussion*: This research indicates that PSO has significant protective benefits against IR injury in the lower extremities. PSO mitigated tissue damage and maintained mitochondrial integrity by addressing oxidative stress, inflammation, and apoptotic pathways. Particularly, high-dose PSO yielded more substantial enhancements in these processes and exhibited outcomes most comparable to the control group in biochemical, histological, and immunohistochemical investigations. These findings underscore the potential of PSO as an efficacious natural treatment agent for IR injury. Nevertheless, additional research is required to articulate this definitively.

## 1. Introduction

Peripheral artery disease is a condition resulting from the ischemia of vessels supplying the lower extremities, progressing to ischemic rest discomfort, foot ulceration, and potential limb loss in the future [1,2]. The severity of ischemia is dictated by the critical ischemia duration, which is the maximum time that tissue can endure ischemia while remaining viable. The duration ranges from 4 to 6 h in nerve tissues and 6 to 8 h in tissue of muscles [3]. While prompt revascularization is favored to avoid surpassing the critical ischemia threshold, reperfusion of the ischemic muscle carries the risk of immediate tissue injury [4]. Reactive oxygen species (ROS), which contribute to cellular damage, are generated during ischemia, and upon reperfusion, these ROS disseminate from the affected cells to remote organs, simultaneously inducing further ROS production and resulting in oxidative stress and lipid peroxidation [5]. Inflammatory mediators emitted from injured cells further exacerbate tissue damage [6]. Furthermore, the antioxidant pathway responsible for mitigating cellular damage is compromised [7]. Therefore, to prevent damage caused by IR, reducing ROS production, strengthening the antioxidant defense system, reversing cellular oxidative stress, and lipid peroxidation will prevent cellular damage. Phytochemicals have recently been employed to mitigate these effects and obstruct the harm pathway in IR [8].

Pomegranate (*Punica granatum* L.) is regarded as a polyphenol-rich food and is extensively utilized in traditional medicine. The antioxidant efficacy of pomegranate juice is acknowledged as superior to that of other fruits [9]. Pomegranate seed oil (PSO) exhibits potent antioxidant, anti-inflammatory, anti-apoptotic, and anticarcinogenic properties [10,11,12,13,14]. Their characteristics stem from the elevated levels of bioactive substances, including polyphenols, fatty acids, and tocopherols [11,12,13].

This study sought to examine the possible therapeutic consequences of PSO on lower extremity IR injury and to assess the mechanisms underlying these effects through biochemical, histological, and immunohistochemical investigations. The objective was to ascertain the appropriate therapy dosage by comparing the beneficial effects of low and high dosages of PSO. We concentrated on diminishing reactive oxygen species (ROS) production by replenishing oxygen in the tissues, prioritizing the application of PSO prior to reperfusion. Throughout this process, we monitored the activity of the inducible transcription factor subfamily NF-κB, which governs apoptotic mechanisms, oxidant/antioxidant responses, inflammatory reactions, and other genes implicated in various aspects of these processes. Consequently, we proposed that PSO may serve as a treatment method by elucidating the potential mechanism of action in the cellular life and death processes inside lower extremity IR.

## 2. Materials and Methods

### 2.1. Study Groups and Chemicals

The necessary number of samples for the study was determined using the effect size for the One-Way Analysis of Variance, computed as n^2^ = 0.30 and f = 0.67, with 8 rats per group, yielding 80% statistical power and a 95% confidence level, resulting in a total of 32 rats. The study’s sample size was determined using the Gpower 3.1 software. The research involved 32 male Wistar albino rats, each weighing between 200 and 250 g, provided by the Gazi University Experimental Animals Laboratory Center. The animals were maintained in a conventional laboratory setting featuring a cycle of 12 h of light and darkness, 45–65% relative humidity, and a room temperature of 21 ± 2 °C. Animals were given a routine pellet diet and unrestricted access to water.

A total of 32 animals were randomly assigned to four groups. The groups were determined as control (C), ischemia–reperfusion (IR), low-dose PSO (IR + LD, 0.15 mL/kg), and high-dose PSO (IR + HD, 0.30 mL/kg). Cold pressed PSO (Cas Number: 84961-57-9) was acquired from TROİLS Oil Industry and Trade Inc. (Antalya, Turkey). According to TS EN ISO 12966-2-4 (GC-MS/FID)/27022020/7 analysis, the composition includes 3–4% palmitic, 2–3% stearic, 6–7% oleic, 6–7% linoleic, 0–1% arachidic, 0–3% 11-eicosenoic, and 79–84% punicic.

### 2.2. Experimental Procedure

At the initiation of the process of experimentation, all rats were sedated with ketamine hydrochloride (50 mg/kg), (Ketalar^®^ Flask, Parke-Davis, Detroit, MI, USA) as well as xylazine hydrochloride (10 mg/kg), (Alfazyne (Belgaum, India), 2%, Ege Vet) [15].

For Group C, for which *n* = 8, 0.30 mL/kg of physiological serum was delivered intraperitoneally without causing ischemia in the skeletal muscles of the lower extremities. Blood and skeletal muscle from the lower extremities were obtained under anesthesia four hours after the surgery.

For IR, for which *n* = 8, ischemia was generated in the lower limb skeletal muscle of the rats by placing a crossed clamp on the infrarenal aorta for 120 min post-laparotomy. After 120 min of ischemia occurred, the blood vessel clamp was released [16].

For IR + LD, for which *n* = 8, following laparotomy, a temporary clamp was positioned on the infrarenal aorta for a duration of 120 min, inducing ischemia in the skeletal muscle of the lower extremities. After 120 min of ischemia, the vascular clamp was released, and promptly thereafter, a single dosage of 0.15 mL/kg PSO was delivered intraperitoneally at the onset of reperfusion [17].

For IR + HD, for which *n* = 8, following laparotomy, a temporary clamp was applied to the infrarenal aorta for 120 min, inducing ischemia in the lower extremity skeletal muscle. After 120 min of ischemia, the blood vessel clamp was released, and immediately upon the onset of reperfusion, 0.30 mL/kg of intraperitoneal PSO was delivered [18].

Following reperfusion, samples of blood were obtained from the aorta in groups IR, IR + LD, and IR + HD for biochemical analysis, and gastrocnemius muscle was removed for histological and immunohistochemical examinations.

### 2.3. Biochemical Evaluation

After centrifuging (Hettich Mikro 200R^®^, Tuttlingen, Germany) at 3000 rpm for 10 min, the remaining serum (supernatant) was left in the epend-off microcentrifuge (Eppendorf tube) tube and saved at −80 °C until the day of the study. TOS, TAS, and OSI, which is a proportional comparison of these, were measured from these samples. These measurements were taken at the Atılım University Faculty of Medicine, Physiology Laboratory.

#### 2.3.1. Serum Total Oxidant Status

The TOS kit from (RelAssay Diagnostic^®^, Gaziantep, Türkiye) was utilized. To measure TOS, 500 mL of reagent 1 was combined with 75 mL pertaining to the sample according to the kit protocol, and the absorption was recorded at 530 nm using a spectrophotometer (NanoDrop^®^ ND-1000, Wilmington, DE, USA) (A1). Reagent 2, which is 25 mL of pro-chromogen was incorporated into the mix. The tube was sealed with parafilm and incubated (Şimşek Laborteknik^®^, Ankara, Türkiye) at 37 °C for a duration of 5 min. Post incubation, absorbance was determined at 530 nm (A2). The kit supplied a standardized solution with a concentration of 10 µmol/L of hydrogen peroxide equivalent in each liter for accurate assessment. Initial and subsequent assessment were conducted thrice, and the mean was calculated. The alteration in absorbance (∆Abs) was determined by subtracting the initial value (A1) from the subsequent value (A2). TOSs were ascertained using the following formula, which is in the kit and expressed in mmol H_2_O_2_ Eq/L.

TOS = [(∆Abs sample)/(∆Abs standard)] × Standard Concentration (10 µmol/L) [19].

#### 2.3.2. Serum Total Antioxidant Status

The TAS kit (RelAssay Diagnostic^®^, Gaziantep, Türkiye) was utilized. To measure TAS, 500 mL of reagent 1 and 30 mL of sample were combined as specified in the procedure and measurement was performed at 660 nm with a spectrophotometer (NanoDrop^®^ ND-1000, Wilmington, DE, USA) (A1). An amount of 75 mL of reagent 2 (colored 2,2-azino-bis-3-ethylbenzothiazoline-6-sulfonic acid) and (ABTS) solution was added to the tube mix. The tube was covered with parafilm and incubated (Şimşek Laborteknik^®^, Ankara, Türkiye) at 37 °C for 5 min. Post-incubation measurement was made at 660 nm (A2). The value of standards, Trolox Eq solution at a concentration of 1 mmol/L, was utilized instead of the sample. The evaluations were made and the norm was taken. Change (∆Abs) was measured by subtracting the first value (A1) from the second value (A2). TASs were determined with the kit directions and stated as mmol Trolox Eq/L.

TAS = [(∆Abs H_2_O − ∆Abs Sample)/(∆Abs H_2_O − ∆Abs Standard)] [19].

#### 2.3.3. Oxidative Stress Index

The OSI is assessed as a marker of ROS imbalance. It is defined as the percentage of the ratio of TOS to TAS. During the measurement of OSI values for the samples, TASs are converted to μmol units. Calculations were expressed as Arbitrary Units (AU).

[OSI (Arbitrary 34 Unit) = TOS (μmol H_2_O_2_ Equivalent/L)/TAS (mmol Trolox Equivalent/L) × 100] [19].

### 2.4. Histopathological and Immunohistochemical Analysis

Gastrocnemius muscle specimens were promptly immersed in 10% neutral buffered formalin solution. After 48 h of fixation, samples were processed and embedded in paraffin. Sections with a thickness of five micrometers were cut from paraffin blocks utilizing a microtome (Leica RM2245, Nussloch, Germany). Consecutive tissue sections were stained with hematoxylin and eosin (H&E) for the evaluation of histopathological alterations and labeled with antibodies against TNF-α, NF-κB, CYT C, and CASP3 to examine the impact of PSO on inflammatory and apoptotic mechanisms. All prepared specimens were assessed utilizing a microscope (Leica DM 4000B, Nussloch, Germany) equipped with a computer, and images were captured utilizing the Leica LAS V4.9 software.

H&E-stained specimens were observed at 200× and 400× magnifications, and the histological damage score of samples was evaluated based on two criteria: the disorganization and degeneration of muscle fibers (0: Normal, 1: Mild, 2: Moderate, 3: Severe), and inflammatory cell infiltration (0: Normal, 1: Mild, 2: Moderate, 3: Severe). The combined value of these two scores, which ranged from 0 to 6, was established as the extent of injury to muscles [20].

For immunohistochemical examination, polyclonal primary antibodies were utilized: anti-TNF-α (1:100 dilution, bs-10802R, Bioss, Woburn, MA, USA), anti-NF-κB p65 (1:100 dilution, bs-0465R, Bioss, USA), anti-CYT C (1:100 dilution, bs-0013R, Bioss, USA), and anti-CASP3 (1:100 dilution, bs-0081R, Bioss, Woburn, MA, USA). After deparaffinization and rehydration, tissue sections were immersed in citrate buffer (pH 6.0) for heat-induced antigen retrieval. The activity of endogenous peroxidase in tissue was inhibited by incubating slices in 3% H_2_O_2_. Protein blocking (Thermo Scientific, TA-060-UB, New Castle, DE, USA) was followed by overnight incubation with primary antibodies at 4 °C. Subsequently, incubation with a biotin-conjugated secondary antibody (Thermo Scientific, TP-060-BN, New Castle, DE, USA) was performed and followed by incubation with an avidin-horseradish-peroxidase (HRP) enzyme conjugate (Thermo Scientific, TS-060-HR, USA). Finally, diaminobenzidine (DAB) (Thermo Scientific, TA-060-HDX, Hartselle, AL, USA) was applied to the sections as the chromogen (in color brown) to detect HRP enzyme conjugated to secondary antibody, revealing the immunoreactivity visually. Immunolabeled sections were examined at 400× magnification, and ten randomly selected, non-overlapping areas from each stained specimen were captured to quantify the immunopositive staining intensity for each specific marker using ImageJ software (1.53a; National Institutes of Health) [21]. The immunopositive staining intensity of each TNF-α, CASP3, and CYT C labeled sections was assessed using the H-score, calculated with the formula H-Score = Σ P_i_ (i + 1) [22]. For NF-κB p65 expression, positively stained nuclei were assessed in NF-κB p65 labeled sections, and the “% positive nuclei per high power field (HPF)” value was utilized for comparison.

### 2.5. Statistical Analysis

Statistical analyses were conducted utilizing SPSS version 20.0 (SPSS, IBM Corp., Armonk, NY, USA). The Kolmogorov–Smirnov test was employed to assess the normality of continuous variables. Histopathological parameters that failed to conform to the normal distribution assumption were assessed utilizing the non-parametric Kruskal–Wallis test. One-way ANOVA was utilized to compare immunohistochemical variables and oxidative stress signs (TAS, TOS, and OSI levels) within groups. A *p*-value below 0.05 is deemed statistically significant. Data were presented as mean ± standard deviation (Mean ± SD).

## 3. Results

### 3.1. Biochemical Results

The TAS, TOS, and OSI signs across all groups were discovered to be statistically different from each other (*p* < 0.0001 for all comparisons). The TOS was significantly higher in the IR groups than in the control group (*p* < 0.05). However, TOS was lower in the IR + LD and IR + HD groups compared to the IR in significance (*p* < 0.05). The TAS was markedly lower in the IR and IR + LD groups relative to the control group (*p* < 0.05). The TAS was significantly higher in IR + LD and IR + HD compared to IR (*p* < 0.05). OSI was higher in the IR group than in the control group (*p* < 0.05) in significance, as indicated in Table 1. OSI was decreased in the IR + LD and IR + HD groups compared to the IR group (*p* < 0.05), in significance value as shown in Table 1.

### 3.2. Histopathological and Immunohistochemical Results

Histopathologic analysis of H&E-stained sections from the control group revealed normal muscle fascicles that are formed by bundles of polygonal-shaped muscle fibers in a cross sectional view as well as normal muscle fibers with peripherally located nuclei and cross-striations in a longitudinal sectional view. In the cross sections of tissue samples from the IR group, it was observed that the interstitial space between the muscle fibers within the same fascicle increased significantly due to interstitial edema, and the muscle fibers suffered from varying degrees of degeneration and a loss of integrity due to IR injury. It was also observed that injured cells lost their usual polygonal shape appearing in cross sectional views and rather had round outlines. Additionally, the disruption of cell outlines and defects of cytoplasmic material were also seen in occasional muscle fibers. In the longitudinal sections, fibers with pale, stained sarcoplasm and a loss of cross-striation, as well as an intense inflammatory cell infiltration, were detected (Figure 1).

Statistical comparisons revealed significant differences in the disorganization and degeneration of muscle fibers, inflammatory cell infiltration, and total injury scores across all groups (*p* < 0.0001 for each parameter). Disorganization and degeneration of muscle fibers in the IR and IR + LD groups were significantly greater than in the control group (*p* < 0.05). Disorganization and degeneration of muscle fibers in the IR + LD and IR + HD groups were less pronounced than in the IR group (*p* < 0.05). Inflammatory cell infiltration was significantly greater in the IR, IR + LD, and IR + HD groups compared to the control group (*p* < 0.05). Additionally, infiltration was less pronounced in the IR + LD and IR + HD groups relative to the IR group (*p* < 0.05). The total injury score was significantly elevated in the IR, IR + LD, and IR + HD groups relative to the control group (*p* < 0.05). The scores of the IR + LD and IR + HD groups were significantly lower than that of the IR group (*p* < 0.05). Additionally, the injury score of the IR + HD group was lower than that of the IR + LD group (*p* < 0.05). The aforementioned histological alterations showed improvement in the treatment groups, with more pronounced effects observed in the tissue specimens obtained from the IR + HD group (Table 2).

The immunohistochemical examination revealed an elevation in TNF-α staining intensity in the IR group relative to the control group, along with apparent damage in the muscle fibers. Enhanced NF-κB immunoreactivity with an increased number of immunopositive nuclei was noted in the IR group compared to the control group. Despite a reduction in TNF-α staining in the IR + LD group relative to the IR group, levels were increased in comparison to those in the control group. Low-dose PSO partially diminished TNF-α expression but failed to fully inhibit the inflammatory response. TNF-α staining in the IR + HD group was significantly diminished, approaching levels observed in the control group. NF-κB immunoreactivity was elevated in the IR + HD group relative to the control group, and diminished in comparison to the IR and IR + LD groups. High-dose therapy more effectively inhibited the inflammatory response and NF-κB activation. Differences were noted between the groups concerning TNF-α immunostaining intensity and NF-κB immunopositivity in significance (*p* = 0.003, *p* < 0.001, respectively). The TNF-α immunostaining intensity in the IR and IR + LD groups was significantly elevated as opposed to the control group (*p* < 0.05). However, it was lower in the IR + HD group compared to both the IR and IR + LD groups (*p* < 0.05). NF-κB immunopositivity was higher in the IR and IR + LD groups compared to the control group; however, it decreased in the IR + LD and IR + HD groups relative to the IR group in significance (*p* < 0.05), (Table 3), (Figure 2).

The IR group exhibited elevated staining intensity for both CYT C and CASP3 in comparison to the control group. A reduction in CYT C and CASP3 immunoreactivity was noted in the IR + LD group relative to the IR group; nonetheless, expression levels were still elevated compared to the control group. This suggests that low-dose treatment partially inhibited the mitochondrial-induced apoptosis process, although apoptosis could not be entirely averted. Reduced expression of CYT C and CASP3 limited the apoptosis response to low-dose treatment. Expressions of CYT C and CASP3 were reduced in the IR + HD group, approaching levels similar to those of the control. The reduced immunoreactivity in CYT C indicates that mitochondrial membrane integrity was maintained and apoptosis signaling was suppressed. The marked reduction in CASP3 expression was observed in the high-dose treatment. The immunostaining intensity of CTY C and CASP3 showed statistically significant differences among the groups (*p* < 0.001, *p* = 0.005). The intensity of CTY C immunostaining was significantly greater in the IR, IR + LD, and IR + HD groups compared to the control group (*p* < 0.05). Furthermore, the CTY C immunostaining intensity was found to be lower in the IR + LD and IR + HD groups compared to the IR group (*p* < 0.05). The immunostaining intensity of CASP3 was significantly higher in the IR group compared to the control group; however, it was lower in the IR + HD group than in both the IR and IR + LD groups (*p* < 0.05), (Table 3, Figure 3).

## 4. Discussion

This study illustrates the protective benefits of varying dosages of PSO against oxidative, inflammatory, and apoptosis-related cellular damage induced by the lower extremity IR model, assessed biochemically, histopathologically, and immunohistochemically. Our results indicate that PSO, particularly at elevated doses, inhibits the primary pathophysiological mechanisms of IR damage and preserves the tissue’s histological integrity. Research indicates that reactive oxygen species (ROS) are the primary contributors to IR damage. Oxidative stress reasoned by reactive oxygen species (ROS) triggers the peroxidation of lipids, proteins, as well as nucleic acids, occurring in DNA damage and mitochondrial dysfunction [23]. Our investigation demonstrates that the elevated TOSs and diminished TASs in the IR, relative to the control (Table 1), signify an accumulation of ROS in the tissue and insufficient antioxidant defense mechanisms. The elevation of OSI values in the IR group (Table 1) indicates that the tissue is experiencing significant oxidative stress. Both low- and high-dose PSO treatments reduced TOSs and elevated TASs; however, high-dose PSO yielded a more pronounced enhancement in oxidative stress indicators (Table 1). Our data align with the findings of Bihamta et al. (2017) that PSO diminishes ROS generation and inhibits lipid peroxidation [24]. The protective action of PSO is believed to stem from the enhancement of antioxidant defense systems by inhibiting ROS production via its constituents. The research by Gutierrrez et al. (2021) demonstrated that the significant antioxidant potential of PSO is primarily attributed to punicic acid, a principal component of the PSO utilized in our study [25]. The data indicate that PSO is an effective drug that enhances antioxidant defense against IR damage and mitigates cellular damage resulting from oxidative stress.

Proinflammatory cytokines are widely recognized as primary instigators of cellular damage in lower extremity IR injury [26]. Our analysis revealed a considerable elevation in TNF-α and NF-κB expressions in the IR group, signifying the severity of the inflammatory response post-reperfusion and the exacerbation of tissue inflammation (Table 3, Figure 2). Notable disarray, degeneration, and significant inflammatory cell infiltration were evident in the muscle fibers of the IR (Table 2, Figure 1). These data indicate that the inflammatory response attains levels capable of inducing structural degradation in muscle tissue [27]. NF-κB, a transcription factor activated by reactive oxygen species and proinflammatory cytokines, is pivotal in IR injury [28].

Our data indicate that PSO exhibits an anti-inflammatory impact, as previously demonstrated in a sepsis model [10]. Nonetheless, low-dose PSO treatment failed to fully inhibit inflammation, despite a reduction in TNF-α and NF-κB expressions (Table 3, Figure 2). Conversely, high-dose PSO treatment markedly diminished TNF-α and NF-κB expressions and more successfully inhibited inflammation in comparison to low doses. This indicates that PSO exerts a significant inhibitory influence on inflammatory processes, particularly at elevated dosages. Concurrently with our results, Koyuncu et al. (2024) demonstrated that it inhibited inflammation; additionally, they found that 0.4 and 0.8 mL/kg PSO decreased NF-κB activation in liver and kidney tissues inside a colitis model [14]. The findings highlight the potential of PSO in mitigating inflammation, with its effects on various tissues potentially varying according to the dosage. Consequently, extensive research is required to thoroughly investigate the anti-inflammatory properties of PSO and to establish the appropriate dosage range across various models.

Apoptosis serves as a fundamental mechanism for cell loss in ischemia–reperfusion tissue injury [29,30]. Apoptosis can be initiated via two separate routes. The intrinsic pathway relies on mitochondria and is triggered by reactive oxygen species (ROS), whereas the extrinsic pathway is influenced by inflammatory molecules like TNF-α. Furthermore, TNF-α plays a role in the intrinsic pathway by indirectly stimulating ROS production through NADPH oxidase [3,31]. Phosphorylation of the TNF receptor results in the translocation of NF-κB to the nucleus, facilitating the transcription of various genes, including cytokines, chemokines, and other inflammatory mediators [32]. Consequently, the lower extremity has been proposed as a potential target for therapeutic intervention in IR [33]. Our study demonstrated that both low and high doses of PSO decreased NF-κB activity, aligning with the identified therapeutic target (Table 3, Figure 2). These findings indicate that PSO may modulate intrinsic and extrinsic apoptosis pathways in lower extremity IR damage, potentially offering tissue-protective effects, especially through the suppression of NF-κB activity.

Apoptosis is a multifaceted process governed by the equilibrium between positive and negative regulators. This equilibrium dictates whether the cell will persist or experience programmed cell death. Antiapoptotic proteins (Bcl-2 and Bcl-xL) enhance cell survival, while proapoptotic proteins (Bax, Bad, Bak, and Bid) initiate the apoptosis process [34]. Bcl-2 specifically binds to Bax, inhibiting the creation of mitochondrial pores that facilitate the discharge of hazardous chemicals and thereby averting apoptosis. The overexpression of Bax and the downregulation of Bcl-2 through p53 induce the release of CYT C from the mitochondria and activate CASP3, which plays a crucial role in the effector phase of apoptosis [34]. This mechanism results in enhanced membrane permeability, DNA fragmentation, and cellular apoptosis [35]. Lopera et al. (2024) have shown that the mechanism of action of the herbal drug Buoverdia in the cerebral ischemia–reperfusion model involves TLR4/NF-κB and caspase-3/Bax/Bcl-2 [36]. Furthermore, Ali et al. (2024) have shown that the apoptotic mechanism of pomegranate peel extract in testicular injury operates via CASP3 [37].

In our investigation, the expressions of CYT C and CASP3 were markedly elevated in the IR group relative to the control group (Table 3, Figure 3). The immunohistochemistry detection of CYT C outside of the mitochondria signifies enhanced mitochondrial membrane permeability and elevated ROS levels during reperfusion, alongside the compromise of mitochondrial membrane integrity due to proinflammatory TNF-α influence (Figure 3). The data indicate that mitochondrial membrane integrity was compromised and apoptosis was initiated due to elevated levels of ROS and proinflammatory TNF-α during reperfusion. Low-dose PSO therapy resulted in a reduction in CYT C and CASP3 expressions; nevertheless, it could not achieve the total inhibition of apoptosis (Table 3, Figure 3). High-dose PSO treatment markedly reduced CYT C and CASP3 levels (Table 3, Figure 3), and dramatically diminished CASP3 levels in comparison to low dosages (Table 3). These findings indicate that PSO preserves mitochondrial membrane integrity and efficiently inhibits the advancement of apoptosis. The data indicate that the anti-apoptotic activity of PSO escalates with dosage and positively influences mitochondrial stability.

This study investigated the impact of different PSO concentrations on inflammatory processes, oxidative stress, and apoptosis in a lower limb IR model, utilizing biochemical and histological methods. The effects of PSO were analyzed in a dose-dependent manner; however, the limited amount of studies in the literature has covered a wide range of parameters, and a thorough comparison of the effects of both doses of PSO was included. The integration of immunohistochemical findings with biochemical analyses to elucidate the pathophysiological process of IR injury improved the reliability and depth of the results.

Nonetheless, this study presents certain limitations. The study assessed IR injury during the acute phase, without examining the chronic effects of PSO or its role in long-term tissue healing. A further limitation is that the effects of PSO were confined to the lower extremity IR model; exploring its effects in various tissue and organ models may enhance the findings.

In the lower extremity IR model, ROS is increased due to enhanced TOS and the enhanced OSI, coupled with a decrease in TAS. The stimulation of the NF-κB signaling pathway by TNF-α exacerbates inflammation. The synergistic effect of these pathways results in mitochondrial membrane impairment, prompting CYT C release and CASP 3 activation, ultimately leading to apoptosis. PSO exhibits significant antioxidant, anti-inflammatory, and anti-apoptotic characteristics, alleviating tissue damage caused by lower extremity ischemia–reperfusion injury. PSO applied at low and high doses (0.15 mL/kg and 0.30 mL/kg) reverses the TAS/TOS pathway and suppresses TNF-α-mediated NF-κB activation. Apoptosis is obstructed by the inhibition of CYT C and CASP 3 activation in the tissue. Consequently, PSO may exhibit therapeutic efficacy in the lower extremity IR model via these pathways.

## 5. Conclusions

This study illustrates that PSO has significant protective effects against oxidative, inflammatory, and apoptotic damage resulting from lower extremity ischemia–reperfusion (IR) injury. PSO enhanced antioxidant defenses by diminishing ROS production, attenuated the expression of proinflammatory cytokines, and prevented apoptotic processes by maintaining mitochondrial membrane integrity. High-dose PSO offered superior protection against IR injury vs. to low-dose PSO. The encouraging outcomes of PSO indicate its potential as a natural therapeutic agent for IR injury. Nevertheless, additional study is necessary, particularly regarding long-term effects and effective dosing regimens, to implement these findings in clinical practice. This work establishes a significant foundation for assessing the therapeutic efficacy of PSO in IR-induced disorders.

## Figures and Tables

**Figure 1 medicina-61-00212-f001:**
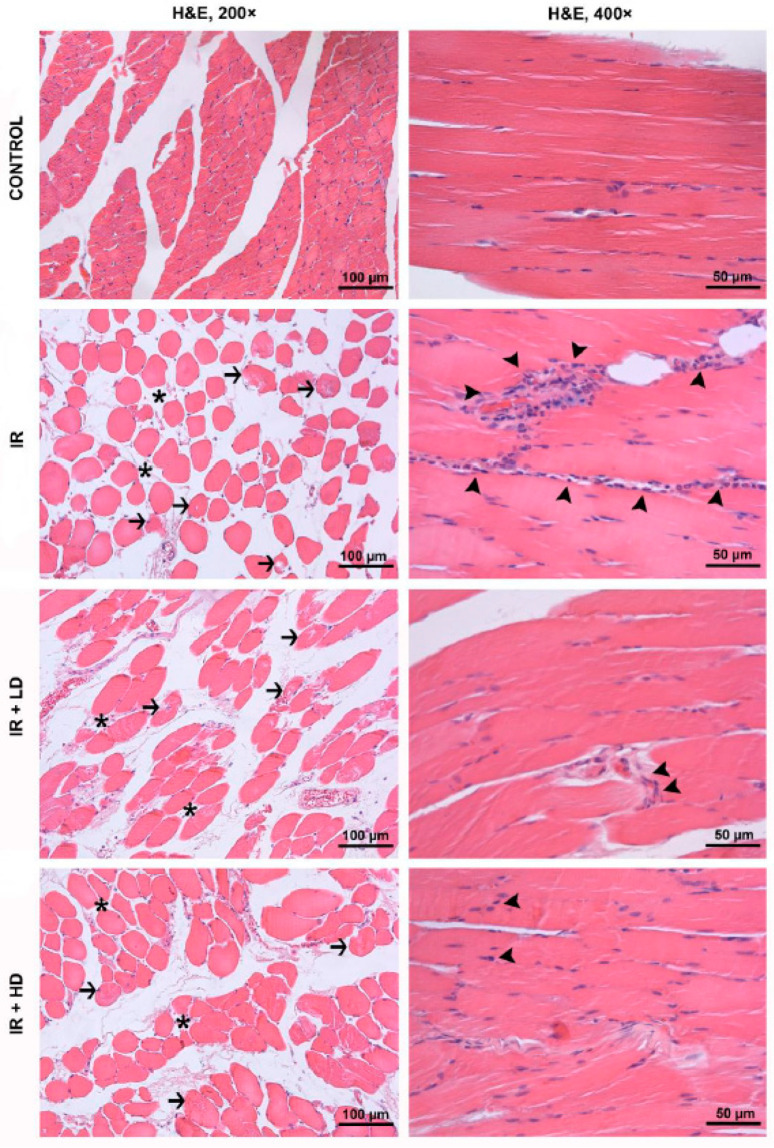
H&E-stained skeletal muscle sections were captured under a 200× in cross sectional plane and 400× magnifications in a longitudinal sectional plane. Black arrowheads: inflammatory cell infiltration; black arrows: muscle fibers exhibiting varying degrees of injury; asterisk: increased interstitial space between muscle fibers due to interstitial edema. IR: ischemia–reperfusion group; IR + LD: low-dose pomegranate seed oil-treated group; IR + HD: high-dose pomegranate seed oil-treated group; H&E: hematoxylin and eosin.

**Figure 2 medicina-61-00212-f002:**
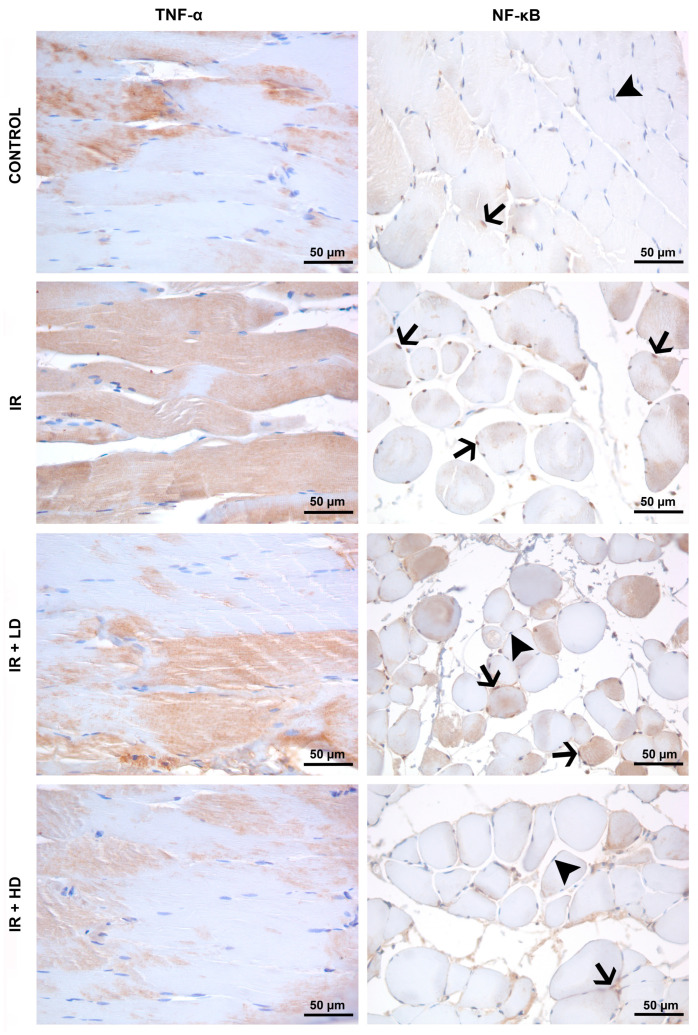
Micrographs representing the TNF-α and NF-κB immunostainings of the groups. Immunopositivity was revealed visually by DAB (in color brown) staining for both TNF-α and NF-κB. Since the nuclear translocation of NF-κB shows its activation state during inflammation, only nuclear positive staining (arrows) rather than cytoplasmic staining was considered for immunoreactivity. Unstained nuclei were indicated by arrowheads. IR: ischemia–reperfusion group; IR + LD: low-dose pomegranate seed oil-treated group; IR + HD: high-dose pomegranate seed oil-treated group. Magnification is 400× for all images.

**Figure 3 medicina-61-00212-f003:**
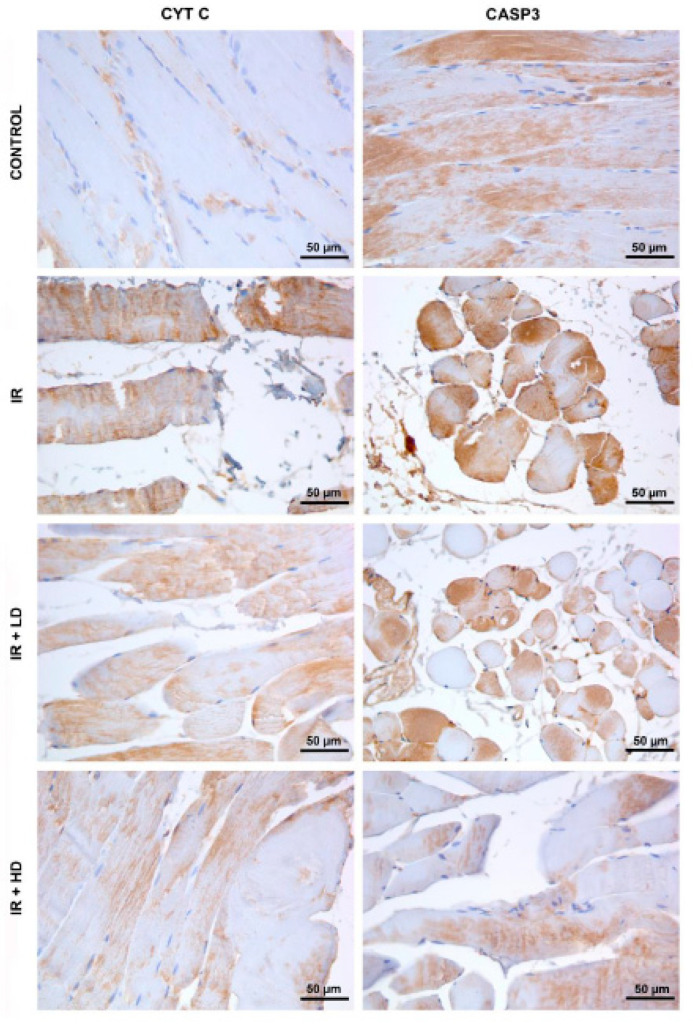
Micrographs representing the CYT C and CASP3 immunostainings of the groups. Immunopositivity was revealed visually by DAB (brown) staining for both CYT C and CASP3. IR: ischemia–reperfusion group; IR + LD; low-dose pomegranate seed oil-treated group; IR + HD; high-dose pomegranate seed oil-treated group. Magnification is 400× for all images.

**Table 1 medicina-61-00212-t001:** Biochemical analysis of serum [Mean ± SD].

	Group C (n = 8)	Group IR (n = 8)	Group IR + LD (n = 8)	Group IR + HD (n = 8)	*p* **
TOS (µmol/L)	6.07 ± 2.65	26.36 ± 9.18 *	12.13 ± 3.49 &	8.14 ± 3.40 &	<0.0001
TAS (mmol/L)	1.17 ± 0.17	0.64 ± 0.07 *	0.89 ± 0.09 *, &	1.10 ± 0.32 &	<0.0001
OSI	5.72 ± 2.98	43.30 ± 13.45 *	13.57 ± 3.64 &	8.13 ± 2.74 &	<0.0001

*p* **: ANOVA test with significance level of *p* < 0.05, * *p* < 0.05 compared with Group C, and & *p* < 0.05 compared with Group IR.

**Table 2 medicina-61-00212-t002:** Histopathological analysis of skeletal muscle tissue [Median (IQR)].

	Group C (n = 8)	Group IR (n = 8)	Group IR + LD (n = 8)	Group IR + HD (n = 8)	*p* **
Disorganization and degeneration of the muscle fibers	1.00 (0–1)	2.50 (2–3) *	1.50 (1–1) *, &	1.00 (0.75–1.25) &	<0.0001
Inflammatory cell infiltration	0.00 (0–0)	2.50 (2–3) *	2.00 (1–2) *, &	1.00 (0.75–1.25) *, &	<0.0001
Total injury score	1.00 (0–1)	5.00 (4–5.25) *	3.50 (2–4) *, &	2.00 (1.5–3) *, &, +	<0.0001

*p* **: Kruskal–Wallis test with significance level of *p* < 0.05, * *p* < 0.05 compared with group C, & *p* < 0.05: compared with group IR, and + *p* < 0.05 compared with group IR + LD.

**Table 3 medicina-61-00212-t003:** Immunohistochemical analysis of skeletal muscle tissue [Mean ± SD].

	Group C (n = 8)	Group IR (n = 8)	Group IR + LD (n = 8)	Group IR + HD (n = 8)	*p* **
TNF-α immunostaining intensity	123.10 ± 16.94	145.41 ± 15.80 *	141.05 ± 14.44 *	110.89 ± 6.83 &, +	0.003
% NF-κB immunopositivity	29.39 ± 7.67	60.45 ± 6.74 *	39.95 ± 9.75 *,&	37.32 ± 3.72 &	<0.001
CYT C immunostaining intensity	115.66 ± 5.34	160.59 ± 13.39 *	136.55 ± 18.46 * &	134.42 ± 16.83 *, &	<0.001
CASP3 immunostaining intensity	147.62 ± 31.47	193.24 ± 17.2 *	170.20 ± 33.79	132.45 ± 10.29 &, +	0.005

*p* **: ANOVA test with significance level of *p* < 0.05, * *p* < 0.05 compared with group C, & *p* < 0.05 compared with group IR, and + *p* < 0.05 compared with group IR + LD.

## Data Availability

All data generated or analyzed during this study are included in this published article.

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
