# Peer review of "Effects of Pomegranate Seed Oil on Lower Extremity Ischemia-Reperfusion Damage: Insights into Oxidative Stress, Inflammation, and Cell Death"

_medicina, 2025, doi:10.3390/medicina61020212_

Round 1
Reviewer 1 Report
Comments and Suggestions for Authors
The paper entitled “Prevention of Lower Extremity Ischemia-Reperfusion Damage with Pomegranate Seed Oil: The Role of Oxidative Stress, Inflammation, and Cell Death” described a novel and interesting therapeutic approach to the use of Pomegranate Seed Oil in the I/R damage in the lower extremities. The study is well-designed, and the results could have a great clinical impact. Methodology is appropriate for determining goals and references are up-to-date.
Although it is obvious that paper deserves attention, there are some corrections to be made:
Results
Missing Table 1, table 2 and Table 3. Please add tables in result section
Discussion
Figure 4 should not stand in the discussion section. Figure 4 moves from the discussion section and describes the mechanism of action of Pomegranate Seed Oil.

Author Response
Dear Editor,
We thank the reviewers for their valuable opinions and contributions to our study. We carefully evaluated the valuable reviewer opinions and made the necessary changes to our manuscript according to these suggestions. We indicate the changes made below.
Reviewer1
Comment:
Missing Table 1, Table 2, and Table 3. Please add tables in the results section.
Figure 4 should not stand in the discussion section. Figure 4 moves from the discussion section and describes the mechanism of action of Pomegranate Seed Oil.
Response:
We appreciate the reviewer's constructive criticism. The “Tables” file contains Table 1, Table 2, and Table 3, including their titles and contents, for your examination. In accordance with the suggestion, Figure 4 has been eliminated from the discussion section, and a paragraph elucidating the technique has been incorporated and highlighted in black font for clarity. Furthermore, Figure 4 has been eliminated from the "Figures" file.
Reviewer 2 Report
Comments and Suggestions for Authors
The manuscript entitled «Prevention of Lower Extremity Ischemia-Reperfusion Damage with Pomegranate Seed Oil: The Role of Oxidative Stress, In-flammation, and Cell Death» authored by Ümmü GülÅŸen Bozok, Aydan İremnur Ergörün, AyÅŸegül Küçük, Zeynep Yığman, Ali DoÄŸan Dursun and Mustafa Arslan has scientific novelty and practical significance. However, there are some minor issues that must be solved before this manuscript can be published:
1. Authors must indicate the positive staining for inflammation markers by arrows or note in the footnotes in Figures 2 and 3 since many readers may not know the details of immunohistochemical staining.
2. It is a bit unclear which areas were used to prepare serial sections and further staining? In addition, did the authors used different areas for TNF, NF-kB, CYT C and CASP 3 staining?
Author Response
Dear Editor,
We thank the reviewers for their valuable opinions and contributions to our study. We carefully evaluated the valuable reviewer opinions and made the necessary changes to our manuscript according to these suggestions. We indicate the changes made below.
Reviewer2
Comments:
- Authors must indicate the positive staining for inflammation markers by arrows or note in the footnotes in Figures 2 and 3 since many readers may not know the details of immunohistochemical staining.
- It is a bit unclear which areas were used to prepare serial sections and further staining. In addition, did the authors use different areas for TNF, NF-κB, CYT C, and CASP3 staining?
Response:
1. Immunoreactivity namely the antigen-antibody binding was revealed visually by using DAB chromogen (brown colored) which is used to detect HRP enzyme conjugated to secondary antibody. Therefore, brown staining indicates the immunoreaction positivity. Since only nuclear staining (in color brown) was considered for NF-κB activation state during inflammation, only nuclear positive staining rather than cytoplasmic staining considered for immunoreactivity. Thus, only the images representing the NF-κB immunostaining were labelled with arrows (showing the positively stained nuclei) and arrowheads (showing the unstained nuclei). Also, it is stated in both material-method section and figure legends that the immunopositivity was revealed visually by DAB (brown) staining for all TNF, NF-κB, CYT C, and CASP3 immunostainings.
- Gastrocnemius muscle specimen were cut into two parts from the midline. Following the completion of tissue processing procedures, these two pieces were embedded in the same paraffin block just adjacent but orthogonal to each other, in a way that we obtain the muscle sections in both longitudinal and cross-sectional planes. Therefore, in each five micrometers sections cut from each muscle specimen by using a microtome, involves both longitudinal and cross-sectional profiles of gastrocnemius muscle specimen. TNF-α, NF-κB, CYT C, and CASP3 staining were performed individually in consecutive sections. And photographs of ten randomly chosen fields from each stained sections of each animal (involving both cross and longitudinal profiles of muscle samples) were captured and analyzed for immunostaining intensity. And at the end an image better representing the results was selected.
Reviewer 3 Report
Comments and Suggestions for Authors
1. Table 1 and table 2 were lost. Please provide Table 1 and table 2.
2. Please assay TAS and TOS of skeletal muscle, and provide the related data.
3. Please assay the activities of major antioxidant enzymes and the levels of oxidative makers in skeletal muscle.
4. Please indicate the exact location of skeletal muscle used in histopathological and immunohistochemical analysis. In figure 1 – 3, there are significant differences in the structure of skeletal muscle.
Author Response
Dear Editor,
We thank the reviewers for their valuable opinions and contributions to our study. We carefully evaluated the valuable reviewer opinions and made the necessary changes to our manuscript according to these suggestions. We indicate the changes made below.
Reviewer3
Comments:
- Table 1 and Table 2 were lost. Please provide Table 1 and Table 2.
- Please assay TAS and TOS of skeletal muscle and provide the related data.
- Please assay the activities of major antioxidant enzymes and the levels of oxidative markers in skeletal muscle.
- Please indicate the exact location of skeletal muscle used in histopathological and immunohistochemical analysis. In Figures 1–3, there are significant differences in the structure of skeletal muscle.
Response:
We appreciate the reviewer for their insightful remarks. Tables 1 and 2, along with their respective titles and descriptions, are contained in the "Tables" file for your examination. Furthermore, each table is supplemented by a comprehensive explanation.
- Gastrocnemius muscle specimen were cut into two parts from the midline. Following the completion of tissue processing procedures, these two pieces were embedded in the same paraffin block just adjacent but orthogonal to each other, in a way that we obtain the muscle sections in both longitudinal and cross-sectional planes. Therefore, in each five micrometers sections cut from each muscle specimen by using a microtome, involves both longitudinal and cross-sectional profiles of gastrocnemius muscle specimen. TNF-α, NF-κB, CYT C, and CASP3 staining were performed individually in consecutive sections. And photographs of ten randomly chosen fields from each stained sections of each animal (involving both cross and longitudinal profiles of muscle samples) were captured and analyzed for immunostaining intensity. And at the end an image better representing the results was selected.
Round 2
Reviewer 1 Report
Comments and Suggestions for Authors
Considering that the authors have corrected the manuscript in accordance with the recommendations, the reviewer suggests to the editorial board of the journal accepting the proposed manuscript.